# The effect of early oral postoperative feeding on the recovery of intestinal motility after gastrointestinal surgery: Protocol for a systematic review and meta-analysis

**Federica Canzan**[ID][1]*, **Arianna Caliaro**[2], **Maria Luisa Cavada**[3], **Elisabetta Mezzalira**[1], **Salvatore Paiella**[4], **Elisa Ambrosi**[1]

1 Dipartimento di Diagnostica e Sanità Pubblica, Università degli Studi di Verona, Verona, Italy, 2 Corso di Laurea in Infermieristica, Azienda Ospedaliera Universitaria Integrata, Verona, Italy, 3 Corso di Laurea in Infermieristica, Scuola Provinciale Superiore di Sanità, Bolzano, Italy, 4 Dipartimento di Chirurgia Generale e Pancreatica, Istituto del Pancreas, Università degli Studi di Verona, Verona, Italy

* federica.canzan@univr.it

**Data Availability Statement:** No datasets were generated or analysed during the current study. All

## Abstract

### Background

Given the ever-shorter length of hospital stay after surgical procedures, nowadays it is more important than ever to study interventions that may have an impact on surgical patients' wellbeing. According to the ERAS (Enhanced Recovery After Surgery protocols) program, early feeding must be considered one of the key components to facilitate early recovery while improving outcomes and patients' overall experiences. To date, the international literature has reported that early postoperative feeding compared with traditional (or late) timing is safe; nevertheless, small clinical outcomes effects has been reported, also for recovery of gastrointestinal function. Therefore, the effectiveness of early postoperative feeding to reduce postoperative ileus duration remains still debated.

### Objective

To analyse the effects of early versus delayed oral feeding (liquids and food) on the recovery of intestinal motility after gastrointestinal surgery.

### Search methods

*Pubmed, Embase, Cinahl, Cochrane Central Register of Controlled Trials (CENTRAL), and* the *ClincalTrials.gov* register will be searched to identify the RCTs of interest.

### Study inclusion

Randomized clinical trials (RCTs) comparing the effect of early postoperative versus late oral feeding on major postoperative outcomes after gastrointestinal surgery will be included.

relevant data from this study will be made available upon study completion.

**Funding:** The authors received no specific funding for this work

**Competing interests:** The authors have declared that no competing interests exist

## Data collection and analysis

Two review authors will independently screen titles and abstracts to determine the initially selected studies' inclusion. Any disagreements will be resolved through discussion and consulting a third review author. The research team members will then proceed with the methodological evaluation of the studies and their eligibility for inclusion in the systematic review.

## Introduction

### Description of the condition

The modern meaning of the ileus condition reflects the absent or reduced peristalsis that can be attributed to a normal, prolonged, or pathological response of the gastrointestinal tract. This failure of peristalsis results in the accumulation of gastrointestinal secretions leading to abdominal distention and nausea, possibly vomiting [1].

By this definition, postoperative ileus is a transient cessation of coordinated bowel motility after surgical intervention, which prevents effective transit of intestinal contents or tolerance of oral intake resulting in the presence of nausea, vomiting, and the failure to pass flatus or stool [1]. Furthermore, it may become a source of pain and discomfort for the patient and leading to the consequent onset of impaired nutritional requirements and protein deficiency and, therefore to increased risk of infection, delays in hospital discharge increasing costs [2–4].

In the literature, some authors as Bragg et al. [1] and Holte & Kehlet [5], described how the return of a normal gastrointestinal function after surgery follows, in most cases, three stages: 1) the small bowel recovers between 0 and 24 hours, 2) the stomach between 24 and 48 hours and 3) the colon between 48 and 72 hours. However, other authors like Delaney et al. [6] state that gastric and small bowel activities should return to normal function just a few hours after surgery. As this background, it is still debated how to distinguish between a physiological and a prolonged status of ileus. Moreover, postoperative ileus (POI) is a common response to abdominal surgery, with an estimated incidence between 17% and 80% [7]. The high variability of incidence reported in the literature reflects the absence of an established and shared definition of what can be accepted as a "normal" duration of the intestinal paralysis [1, 3]. Nevertheless, Lee et al. [8] highlighted that postoperative ileus could be considered resolved if the following requirements are concomitantly present: food tolerance, absence of abdominal distension, the passage of flatus, or stools.

Because of the lack of consensus regarding the POI definition, some uncertainties remainy regarding its assessment and therefore regarding which outcomes may be the most appropriate to evaluate. Many different effects have been proposed in the literature as an expression of POI resolution, such as the reappearance of bowel sounds and passage of flatus or bowel movement, tolerance to fluid and food consumption, and reinsertion of the nasogastric tube together with the presence of bowel sound. However, these seem to be poor markers of Ileus recovery [9].

Currently, the most effective approaches for postoperative ileus tend to be focused on prevention rather than treatment. Some interventions, as the adoption of minimally invasive surgery [10], perioperative fluid management strategies based on the "Goal-Directed Therapy" [11], and the use of local epidural anaesthetics in lieu of opioid-based analgesic regimens [12], have been thoroughly studied and have been considered effective for ileus prevention.

One of the five key component of the ERAS (Enhanced Recovery After Surgery protocols) program [13] is enforced early feeding. In current practice, early feeding is reached using different kind of interventions such as chewing gum, [4] coffee consumption [14], sips of water clear fluid, tube feeding and so on, sometimes a combination of these interventions can be proposed [9, 15]. Chapman and colleagues [4] reported that early postoperative feeding compared with traditional (or late) timing was safe and reduced length of hospital stay; nevertheless, they reported small effects in hospital discharge with an uncertain clinical significance, shoving minimal effects on a several measurement of gastrointestinal function. Although early oral feeding is considered by the international literature a common and safe intervention for postoperative ileus prevention, the current evidence regarding early *versus* delayed postoperative feeding does not highlight any significant change regarding the incidence of wound infections, intra-abdominal abscesses, or pneumonia [16]. Therefore, the effectiveness of early postoperative feeding to reduce ileus duration remains debated.

### Description of the intervention

Over the last few years, early postoperative feeding (within 24 hours from surgery) was among the many preventive multimodal interventions introduced in the Enhanced Recovery After Surgery protocols (ERAS). The ERAS has been proven effective in accelerating the post-surgical recovery and reducing hospital length of stay, although the relationship between early feeding and ileus has not yet been directly evaluated. To date, it has been considered safe to resume feeding early in the postoperative period without the risk of negative impacts on the anastomosis dehiscence [17].

### How the intervention might work

Early oral nutrition allows the gastrointestinal system to regain its physiological status and functions more rapidly than parenteral nutrition and specifically stimulates motility, which will allow a faster first passage of flatus and stools [18]. In addition, on the one hand, early oral nutrition would prevent significant metabolic changes that occur within 24 hours of starvation, such as increased insulin resistance [19]; on the other, it would readily provide the fundamental constituents for surgical wound healing.

### Why it is important to perform this review

The postoperative time and day to be considered safe to resume oral intake and the concept of "early feeding" is still discussed and varies widely between surgical contexts. Moreover, there is no shared definition regarding the most appropriate outcomes that need to be used to evaluate postoperative ileus and the most appropriate time and nutrients to administer to stimulate peristalsis.

The intervention "feeding" itself remains vague due to the lack of consensus regarding what is considered feeding (liquids versus solid food, for example) and what is the best approach in terms of types of nutritional components routinely prescribed to the patients to improve intestinal motility, which may vary widely across surgical settings and cultures and therefore being context specific.

Most of the previously published systematic reviews considered only colorectal [9, 20, 21] or lower gastrointestinal surgery [16], or assessed other primary outcomes rather than POI, such as length of stay (LoS) and postoperative complications [23].

The strategies currently used to improve postoperative intestinal motility concerning oral feeding include the administration of sips of water, clear fluids, coffee, restrictions to or combination of certain food types [4].

## Objectives

The main aim of this study is to provide a systematic review of the effects of early *versus* delayed oral feeding (liquids and food) on the recovery of intestinal motility after gastrointestinal surgery.

## Methods

### Study registration

This systematic review has been registered prospectively in the International Prospective Register of Systematic Reviews (PROSPERO) the registration number is CRD42022298777.

The protocol aligns with the PRISMA-P guidelines [22], and the checklist is available in S1 Appendix.

### Criteria for including studies in this review

**Types of studies.** Randomized clinical trials (RCTs) comparing the effect of early postoperative versus late oral feeding on major postoperative outcomes after gastrointestinal surgery will be included. In fact, RCTs are considered to provide the strongest measure of whether an intervention has an effect. Because of the "fast-track" approach, which includes post-operative early feeding among other core components, has been developed in the 1990s, studies published since 1990, will be considered. Moreover, due to the language skills of the research team, studies in English, Italian, and German will be assessed.

**Types of participants.** Participants included in this study will be adult patients (>18 years old) undergoing both elective and emergency gastrointestinal surgery regardless of the type of incision, surgical technique, and type of anesthesia. Patients undergoing bariatric surgery, due to its metabolic nature, appendectomy and proctological surgery will be excluded, as well as gynecological procedures.

**Types of intervention.** The main intervention will be the early postoperative oral feeding with fluids and food. It will be considered "early" if feeding was started within 24 hours from the end of the surgery and in case the food and/or fluids were administered by mouth. We will not include enteral and parenteral feedings.

The control intervention will be "delayed" postoperative feeding, defined as if it started after 24 hours from the end of the surgery.

**Types of outcome measures.** Primary outcome: time to first passage of stool (first passage in days).

Secondary outcomes: time to onset of first flatulence, the first sound, and bowel movement, the onset of gastrointestinal adverse events such as nausea, vomiting, diet intolerance, cramps, distension, and abdominal pain, length of hospital stay, and other adverse events.

### Search methods for identification of studies

*Pubmed, Embase, Cinahl, Cochrane Central Register of Controlled Trials (CENTRAL), and* the *ClincalTrials.gov* register will be searched to identify the RCTs of interest. Additionally, the SIGLE System for Information on Grey Literature (SIGLE) will be consulted to identify further studies or papers not published. The bibliographic references will be revised, and, if necessary, the corresponding authors will be contacted to clarify doubts and consider unpublished data.

**Electronic searches.** The research in the databases will be performed using the terms as both free texts as well as MeSH and EMTREE terms by adopting the search strings specified in S2 Appendix.

## Data collection and analysis

**Selection of studies.** The records identified through the search methods will be transferred in Excel® (Microsoft, 2021) spreadsheets. Two review authors will independently screen titles and abstracts for relevancy in Covidence. Any disagreements will be resolved through discussion and consulting a third review author. The research team members will then proceed with the methodological evaluation of the studies and their eligibility for inclusion in the systematic review.

**Data extraction and management.** For each RCT included, the following data will be extracted using electronic data collection forms in Covidence:

- article references (first author, journal, year)

- study setting

- research methods (study design, study total duration, washout period)

- type of surgery (emergency, or elective surgery)

- participants characteristics (age, gender)

- intervention (main and control)

- study's primary and secondary outcomes

- main results

- free notes

**Risk of bias assessment.** Two independent reviewers will perform the quality and risk of bias assessment of the selected studies using the Cochrane Handbook for Systematic Reviews of Intervention (Higgins, 2011) [23] tool. The risk of bias will be assessed according to the following domains:

- random sequence generation

- allocation concealment

- blinding of participants and personnel

- blinding of outcome assessment

- incomplete outcome data

- selective reporting bias

- other bias.

The risk of bias of each included trial will be classified as high, low, or moderate based on the following criteria:

- low risk: assigned to studies with low risk in all key domains;

- high risk: accorded to studies found with high risk in one or more key domains;

- unclear risk of bias: in case of unclear risk of bias found in one or more key domains.

**Measure of treatment effect.** We will use the mean difference (MD) to assess the treatment effect of the continuous variables, like the primary outcome, with 95% confidence intervals (CIs), if the studies presented the same measurement scales. If the studies do not show homogeneity regarding the choice of the measurement scales, we will recur to standard mean difference

(SMD). We will also consider the frequency of postoperative complications and the adverse events related to the surgery, applying for dichotomous data a risk ratio (RR) with 95% CIs.

**Dealing with missing data.** The authors of the trials will be contacted to retrieve relevant missing data. We will conduct a sensitivity analysis to assess the impact on the overall treatment effects, and we will address in the discussion the impact of the missing data on the overall effects of the systematic review.

**Assessment of heterogeneity.** Clinical heterogeneity will be determined based on the methodology of the studies and the demographic data of each study's participants. As regarding statistical heterogeneity, we will assess it across studies by visual inspection of the plot and applying the $I^2$ statistic with Q statistic test.

We will apply the following thresholds for interpretation [23]:

- 0–40%: low heterogeneity

- 30–60%: moderate heterogeneity

- 50–90%: substantial heterogeneity

- 75–100%: considerable heterogeneity

**Assessment of reporting bias.** A funnel plot will be used to assess reporting bias if more than ten trials will be included for meta-analysis.

**Data synthesis.** We will perform a narrative synthesis of the sample, intervention, and outcome findings. Dichotomous data will be calculated as summary risk ratios (RR) with 95% confidence intervals (CI). We will perform statistical analysis using Review Manager 5.3 software (RevMan 2014) with a fixed-effects model. We will calculate the $I^2$ statistic and P-value from Chi-squared. If the $I^2$ is higher than 50%, the random-effects model will be used, and the reasons for this heterogeneity will be investigated. If no more than ten trials are included for meta-analysis, then we will use funnel plots for reporting bias assessment.

**Subgroup analysis.** We will perform a quantitative meta-analysis of RCTs retrieved in the systematic review. We plan to conduct these subgroup analyses:

- type of intervention: upper vs. lower gastrointestinal tract (proximal or distal to Treitz's ligament, respectively);

- emergency vs. elective surgery;

- cancerous or non-cancerous conditions.

**Summary of evidence.** The quality of evidence for all outcomes will be judged using the Grading of Recommendations Assessment, Development and Evaluation (GRADE) working group methodology [22]. It will be considered 'high', 'moderate', 'low' or 'very low' based on: risk of bias, inconsistency, indirectness, imprecision, publication bias, and additional domains. The results will then be presented.

## Ethics and dissemination

This review will not include private information from individuals and will not affect patients and their rights. For this reason, ethical approval is not indicated. The results of this scientific project will be disseminated through congresses and peer-reviewed publications.

## Conclusion

This meta-analysis is expected to provide objective evidence of the effects of early vs. delayed oral feeding (liquids and food) on the recovery of intestinal motility after gastrointestinal

surgery and on which oral intake should be considered more effective to prevent postoperative ileus according to the different types of surgery.

## Supporting information

**S1 Appendix. The PRISMA-P checklist.**
(DOCX)

**S2 Appendix. The search strategy for PubMed.**
(DOCX)

## Author Contributions

**Conceptualization:** Federica Canzan, Arianna Caliaro, Maria Luisa Cavada, Salvatore Paiella, Elisa Ambrosi.

**Data curation:** Maria Luisa Cavada.

**Methodology:** Federica Canzan, Arianna Caliaro, Maria Luisa Cavada, Elisabetta Mezzalira, Salvatore Paiella, Elisa Ambrosi.

**Software:** Arianna Caliaro, Maria Luisa Cavada, Elisabetta Mezzalira, Salvatore Paiella.

**Supervision:** Federica Canzan, Salvatore Paiella.

**Writing – original draft:** Federica Canzan, Arianna Caliaro, Elisa Ambrosi.

**Writing – review & editing:** Maria Luisa Cavada, Elisabetta Mezzalira, Salvatore Paiella.

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
