## [Decision Letter · Decision Letter 0]

18 Jul 2022

PONE-D-21-41001The effect of early oral postoperative feeding on the recovery of intestinal motility after gastrointestinal surgery: Protocol for a systematic review and meta-analysisPLOS ONE

Dear Dr. Canzan,

Thank you for submitting your manuscript to PLOS ONE. After careful consideration, we feel that it has merit but does not fully meet PLOS ONE’s publication criteria as it currently stands. Therefore, we invite you to submit a revised version of the manuscript that addresses the points raised during the review process.

We look forward to receiving your revised manuscript.

Kind regards,

Alberto Meyer, MD, PhD

Academic Editor

PLOS ONE

Journal Requirements:

Reviewers' comments:

Reviewer's Responses to Questions

**Comments to the Author**

1. Does the manuscript provide a valid rationale for the proposed study, with clearly identified and justified research questions?

Reviewer #1: Yes

Reviewer #2: Partly

2. Is the protocol technically sound and planned in a manner that will lead to a meaningful outcome and allow testing the stated hypotheses?

Reviewer #1: Yes

Reviewer #2: Yes

3. Is the methodology feasible and described in sufficient detail to allow the work to be replicable?

Reviewer #1: Yes

Reviewer #2: Yes

4. Have the authors described where all data underlying the findings will be made available when the study is complete?

Reviewer #1: Yes

Reviewer #2: Yes

5. Is the manuscript presented in an intelligible fashion and written in standard English?

Reviewer #1: Yes

Reviewer #2: Yes

6. Review Comments to the Author

You may also provide optional suggestions and comments to authors that they might find helpful in planning their study.

Reviewer #1: I would recoomend a program like Covidence for systematic review management, with proper blinding of the reviewers

Reviewer #2: Thank you for the opportunity to review this manuscript – I have found the topic interesting and important to the field of gastrointestinal surgery. A protocol for a systematic review and meta-analysis of the impact of the effect of early oral postoperative feeding on the recovery of intestinal motility.

The authors tackle a major challenge in gastrointestinal surgery. They correctly identify the current literature, with the most up-to-date systematic review results including data on the effect of early post-operative feeding. The methodology for the review is sound with minor comments below. My only limitation lies with what this review will add to the literature, given recent relevant systematic reviews that have covered this topic.

Please consider the following comments.

1. Please could the authors explain their reasoning for language and year restriction in line with AMSTAR 2 guidance. Also, an explanation for non-randomised trial exclusion is warranted.

2. I appreciate the potential reasons for why bariatric surgery was excluded, however the protocol would benefit from a transparent explanation to the reader.

7. PLOS authors have the option to publish the peer review history of their article (what does this mean?). If published, this will include your full peer review and any attached files.

Reviewer #1: **Yes: **Anders C Larsen

Reviewer #2: No

---

## [Author Response · Author response to Decision Letter 0]

1 Aug 2022

Dear Reviewers, 

thank you for your feedback. 

We have really appreciated your suggestions. 

Each point has been considered with care and appropriately addressed as indicated below. 

We have introduced the changes required and we have highlighted them with Track Changes.

Thank you again for your attention and consideration. 

Reviewer #1: I would recoomend a program like Covidence for systematic review management, with proper blinding of the reviewers

Thank you for your feedback. Me and my co-authors have discussed it and we have decided to move forward with this suggestion. We added this information in the Data collection and analysis section.

Reviewer #2: Thank you for the opportunity to review this manuscript - I have found the topic interesting and important to the field of gastrointestinal surgery. A protocol for a systematic review and meta-analysis of the impact of the effect of early oral postoperative feeding on the recovery of intestinal motility.

The authors tackle a major challenge in gastrointestinal surgery. They correctly identify the current literature, with the most up-to-date systematic review results including data on the effect of early post-operative feeding. The methodology for the review is sound with minor comments below. 

My only limitation lies with what this review will add to the literature, given recent relevant systematic reviews that have covered this topic.

This is a very interesting point. The reserch group is aware that there are other recent SRs on this topic but limited to colorectal or lower gastrointestinal surgical patients. Moreover, the outcomes were heterogeneus and other primary outcomes rather than post-operative ileus were measured. We added these reasons in the “Why it is important to do this review”section. 

Please consider the following comments.

1. Please could the authors explain their reasoning for language and year restriction in line with AMSTAR 2 guidance. Also, an explanation for non-randomised trial exclusion is warranted.

We added the information requested in the Types of studies paragraph.

2. I appreciate the potential reasons for why bariatric surgery was excluded, however the protocol would benefit from a transparent explanation to the reader.

Thank you for rising this relevant point. We deepen discussed on this issue whitin the reserch group during the protocol development. Thus, we decided to exclude the bariatric surgery due to its metabolic nature that implies different kind of effects and a post-operative care tailored to patient’s needs (PMID: 24194467). We added a statement in the “Types of participants” paragraph.

---

## [Editor Report · Decision Letter 1]

3 Aug 2022

The effect of early oral postoperative feeding on the recovery of intestinal motility after gastrointestinal surgery: Protocol for a systematic review and meta-analysis

PONE-D-21-41001R1

Dear Dr. Canzan,

We’re pleased to inform you that your manuscript has been judged scientifically suitable for publication and will be formally accepted for publication once it meets all outstanding technical requirements.

Kind regards,

Alberto Meyer, MD, PhD

Academic Editor

PLOS ONE

Additional Editor Comments (optional):

Dear Authors,

Thank you for accepting our recommendations for revision and incorporating the relevant changes.

I am satisfied with your response and thus happy to recommend in favour of publication of your study.

Kind regards
---

## [Editor Report · Acceptance letter]

8 Aug 2022

PONE-D-21-41001R1 

The effect of early oral postoperative feeding on the recovery of intestinal motility after gastrointestinal surgery: Protocol for a systematic review and meta-analysis 

Dear Dr. Canzan:

I'm pleased to inform you that your manuscript has been deemed suitable for publication in PLOS ONE. Congratulations! Your manuscript is now with our production department. 

Kind regards, 

on behalf of

Professor Alberto Meyer 

Academic Editor

PLOS ONE